# Rapid Urbanization in Ethiopia: Lakes as Drivers and Its Implication for the Management of Common Pool Resources

**Aklilu Fikresilassie Kabiso** [1,*]**, Eoin O'Neill** [1] **, Finbarr Brereton** [1] **and Wondimu Abeje** [2]

[1] School of Architecture, Planning and Environmental Policy, University College Dublin, D04 V1W8 Dublin, Ireland
[2] Center for Regional and Local Development Studies, Addis Ababa University, Addis Ababa P.O. Box 1176, Ethiopia
\* Correspondence: aklilu.kabiso@ucdconnect.ie

**Abstract:** Recent history has been marked by a shift from rural to urban living. Studies show that urbanization is most prevalent at coastal areas and river basins and these are the locations where most megacities are established. However, in the African context, there is a deficit of research in this area. The focus of studies in the 'urban' field show the expansion of cities towards waterbodies but with little or no attention to the implications of this expansion—'the rural to urban shift'—particularly as they concern lakes as commons in a rapidly urbanizing world, such as African countries and the Global South. Thus, using the case of lakes in Ethiopia, this study explores the trend of urbanization vis-à-vis lakes and its implications for the management of lakes, where historically the Ethiopian urban system has been characterized by settlements on mountain areas as strategic places located far from water bodies, particularly lakes. Using secondary data on population of urban centers and distribution of lakes in Ethiopia, this paper finds that urban centers that are located adjacent to lakes have been growing faster than those cities and towns that are not. The study argues that lakes are an attraction factor for urbanization. Moreover, rapid urban expansion around lakes implies that, in the future, the management of lakes (as common pool resources) critically depends on how urban centers are planned and managed.

**Keywords:** urbanization; cities; lakes; common pool resources

## 1. Introduction

Urbanization is defined as the process of population concentration in a small area on a permanent basis, forming urban centers [1,2]. It occurs mainly due to the movement of people from rural areas to urban areas which in turn results in growth in the size of the urban population and the developed areas followed by other changes in land use, economic activity, and culture. Urbanization, with its very nature of concentrated population and infrastructure, is the source of increased economies of scale, innovation, and knowledge, which in turn promotes economic growth through raising productivity [3–6]. However, unplanned and rapid urbanization can cause considerable negative impacts on the environment [7–10], particularly imposing significant problems on water bodies, including lakes [11–15].

Currently, 55% of the world's population lives in urban areas, which is expected to reach 60% by 2050. Africa's urban population growth rates have been the highest. The continent's urban population was 395 million in 2010 and is projected to reach 1.5 billion in 2050 [16]. However, there are large variations in the patterns of urbanization across African regions. For example, North Africa has a higher proportion of urban population (48%) relative to Sub Saharan Africa (SSA) which is about 33% urbanized. The continent has had the world's fastest annual average urban population growth rate, approaching 4% [17]. Generally, Africa's urban population is still expected to grow at an average annual rate of almost 3% per year and double in approximately 25 years [16].

Globally, fourteen out of seventeen top megacities are located in coastal areas, and over 50% of the world's population lives in river basins, on the banks of rivers such as Ganga, Indus, Mekong, Zambezi, Congo, Niger, Euphrates and Tigris, Jordan, Danube, Rhine, Colorado, and the Amazon [18]. Moreover, over 50% of the world's population lives closer than 3 km to a surface water body [19], which implies increasing settlements around water bodies. A report on world cities by UN Habitat [17] indicates that in the era of this rapid urbanization, the achievement of sustainable development depends on successful management of urban growth. The shift towards a world dominated by urban implies not only a demographic change characterized by the movement of population from one place to another, but also a transformative process that shapes several aspects of development. The dominance of urbanization is also associated with wide-ranging modifications on land use [20], an increase in consumption levels [21], degradation of natural resources [22], ecological degradation and pressure on ecosystem services [23], habitat loss and ecosystem change [24] all of which are causing several social consequences [25].

Meanwhile, the total area occupied by cities is very small, comprising less than 3% of the global terrestrial surface [26]. However, they are having a significant impact on biodiversity [27–29]. Cities have become responsible for 78% of carbon emissions, between 67% and 76% of global energy use, 60% of residential water use, and 76% of wood used for industrial purposes [30]. In addition, cities have a degrading influence on surface water [31] and loss of natural habitat (urban growth's impact on natural habitat which is attributed to the dramatic changes in land surface characteristics, such as soil properties, vegetative cover, and runoff potential) [32]. It also impacts groundwater status, both quality and quantity, and it is adversely affecting its capacity to recharge [7]. Hence, it is not only the size of the land area that is occupied by urban areas that matters, but also the level of the influence that the urban areas have on the environment. In many African countries, cities pose extreme hazards to water quality through pollution due to lack of proper planning and poor solid-waste management [33]. The situation could be more severe under conditions where cities are built and around lakes without proper planning.

In generic terms urban expansion is caused by rural to urban migration, natural urban population growth—the predominance of births over deaths, reclassification of human settlements, and changes in population [34–36]. However, the question arises as to why some cities grow faster than others. What is it about the pattern and size distribution of cities and what are the driving forces of such differences? The factors that determine the pattern of urbanization and city size distribution vary between countries and contexts. In order to manage the growth of cities as well as their adverse effects on the ecosystem, exploring their size distribution is vital [37].

As clearly articulated by Krugman [38], most of the studies in the field of city size distribution are grounded on the neoclassical urban systems theory, which is purely market-oriented, dealing with agglomeration economies of city size; new economic geography, which looks at the effects of the interactions among market size, increasing returns of firms and transportation costs; and views that cities emerge spontaneously or "in a random process". Within these theories, the factors that contribute to the city size distribution include government interventions policies [38]; the nature of agglomeration and policies [39]; and investment and policies that nurture the development of cities [40]. Some studies also argue that it is the ecological factors that determine the distribution of human settlements considering social and economic processes as means of peoples' survival [37].

Zipf, who introduced Zipf's law of the rank-size rule (rank-size rule is a situation where the number of cities whose population exceeds S is proportional to 1/S, and where the largest city in the urban distribution concentrates economic, social, and political power to reach a level of population size that far outstrips other cities in the system. The assumption here is that the cities are one of the complex systems that are situated in an integrated economic system forming a hierarchical power law function), identified factors such as industrial and commercial development, transport, and expansion of administrative organization as key factors [41]. However, the applicability of Zipf's law has been contested due

to the fact that the world economy is scattered across several cities which in turn repositions the distribution of power (political, economic, and social) [42]. Moreover, Lu [43] compared the rate of urban expansion between coastal and inland cities and showed that coastal cities had faster urban growth rates than inland cities; however, the study used physical conditions such as rivers and bridges as key contributing factors with little attention to the population size.

Furthermore, the context also matters; for instance, the pattern of urbanization of dozens of African countries is associated with the exploitation of natural resources, which indicates that urbanization in such countries is likely to have been driven by the income effect of natural resource endowments [44]. Looking at the evolution of urbanization in Ethiopia, prior studies [18,45] argue that urbanization was started in the mountainous highlands. According to these studies, the word "Ketema"—the Amharic (the national language of Ethiopia) name for urban centre, city, or town, serves as an indicator for the evolution of urbanization in the country. It represents a sign on high ground or a military camp or a strategic high ground [18]. However, given the rapid urbanization in Ethiopia in recent decades, this argument of mountain-based urbanization away from waterbodies may no longer hold true and requires further analysis.

*The Effects of Urbanization on Lakes as Common Pool Resources*

The effect of urbanization is seen to be significant on common pool resources (common pool resources are characterized as resources for which the exclusion of users is difficult (referred to as excludability), and the use of such a resource by one user decreases resource benefits for other users which is referred to as subtractability (Ostrom 1990). CPR examples include earth's oceans and atmosphere, fisheries, forests, irrigation systems, pastures, and lakes) as they are highly vulnerable to urbanization and threats attached to it such as pollution and conversion of land use [46–48]. It is also argued that, due to the urbanization processes, the identity of several common resources have already been transformed into other forms of land use [13]; degraded; polluted; and threatened by high rates of privatization of land and conversion of spaces into buildings, especially in urban areas [47]. There is also rapid increase in built-up areas and the decline of the coverage of vegetation around lakes [49], which has resulted in significant decrease of the size of wetland and water bodies [12]. For instance, studies show that over the last 35 years, the water bodies and wetlands around Bahir Dar city of Ethiopia decreased by 76% [12] and the same effect was observed in other cities of the country.

Lakes are among those resources that are managed as common-pool resources in rural areas across the world [50]. According to Rao et al. [51], lakes are defined as inland bodies of fresh or saline waters, appreciable in size (i.e., larger than a pond), and too deep to permit vegetation (excluding submergent vegetation) to take root completely across its expanse. They also have unique characteristics such as long retention-time, complex-response-dynamics, and integrating nature; however, in most cases, lakes are managed under the generic system (an institutional arrangement designed to manage water resources (i.e., rivers, underground water, streams, lakes, etc.) in general) of managing natural resources [52,53]. For instance, in Ethiopia, lakes are managed under the Ministry of Water and Energy which mainly focuses on water supply and energy. The unique characteristics are not only between lakes and other natural resources, but there is also a significant difference among lakes which requires different governance approaches [53]. On the other hand, urban lakes are more vulnerable to urban and human activities than other natural resources [54]. The impacts of urban and human activities include decreases in lake area and pollution or water quality issues [14,15,55].

Given the significant role and responsibility of cities in affecting large scale ecosystems within their surroundings, it is critical to understand how changes in urban land use and governance affect the use of urban ecosystems [56]. The significance of such studies is more important in the developing world as the changes in urban land which are caused by urbanization are rapid and unplanned [57]. Moreover, the studies on understanding

the effects of urbanization on ecosystems have been limited in developing countries in comparison with studies in the developed world [58]. Further, the concepts and efforts towards sustainable urbanization are inclined to local contexts with lack of attention to the conservation of resources that are located beyond the urban centers [59]. This calls for institutional and governance systems that protect CPRs from being negatively affected by the urbanization processes.

Furthermore, most CPR studies have considered several case studies in relation to agricultural activities, forestry, fishery, pastureland, and individuals' interaction with resource units such as fish, irrigation, livestock, and the use of forests by local communities [50,60–63]. However, such studies have given limited attention to lakes.

Cities have significant negative effects on lakes and there is a need for a proper study on the relationship between urbanization and the management of lakes, which is lacking in the context of Africa in general, and Ethiopia in particular. Therefore, by analyzing the distribution of urban centers and lakes in Ethiopia as well as assessing the differences between different categories of cities (urban centers located adjacent to lakes (within the watershed of lakes) and urban centers not adjacent to lakes (located outside watershed of lakes)), this paper explores the relationship between urban expansion and lakes as CPRs. It also provides policy recommendations for managing the CPRs in the context of rapid expansion of urban centers that are around or are adjacent to lakes.

## 2. Method and Materials

### 2.1. The Study Area: Urbanization and Lakes in Ethiopia

With its population estimated to be 115 million (2020), Ethiopia is the second most populous country in Africa. The level of urbanization in Ethiopia was about 5% in the 1950s and only reached 10% in the 1970s [64]. The Central Statistics Authority (CSA) data show that the share of the urban population increased from about 11% in 1984 to 19% in 2014 and reached close to 22% by 2020.

The urban population forecast based on the CSA projections [65] indicates that by 2030 about 30% of the total population in Ethiopia will live in cities, although the Ministry of Urban Development and Construction (MUDCo) contends that this is a conservative estimate. The CSA [66] estimates that Ethiopia will be about 35% urban even before 2025 based on the assumptions that government driven mega projects and other urbanization drivers contribute to urbanization above and beyond migration and natural growth. However, at this point, the 35% of urbanization is infeasible to be achieved by 2025 from its current level (i.e., 22%) [66]. Even considering this level of urbanization, Ethiopia will be one of the least urbanized countries in Africa by 2030. The World Bank's urbanization review report revealed that, as of 2011, the average level of urbanization for Sub-Saharan Africa was about 37% and that of middle-income countries averaged about 50% [67]. Although the level of urbanization in the country is one of the lowest even by the sub-Saharan Africa standards, the World Bank's urbanization review on Ethiopia described the country as the fastest urbanizing country at a rate of 5.4% per year [67]. The UN Habitat's report on the state of African cities also describes Ethiopia as one of the rapidly urbanizing nations with low initial urbanization levels [30]. Estimates based on the CSA's 2017 projection yield that Ethiopia's urban population is expected to add about 11 million more people to its present level in about 10 years with impacts on CPRs, including lakes.

In Ethiopia, studies also show that changes in the distribution of urban centers (by size) are attributed to the changes in political and policy related issues [68]. According to [68], political and policy related issues include a shift from a centralized system of the Dergue government to a decentralized system under the EPRDF government in 1994. The decentralized system empowers regions to expand their urban centers coupled with the national policies that favor urbanization. [68] argue that the contribution of industries (contrary to Weber [69]) and economic development show less contribution to the growth of cities in Ethiopia as compared to other factors that include distance from large urban centers, administrative location, the administrative hierarchy of the urban centers, transport

infrastructure, and policies. Given the rapid growth of urban centers in Ethiopia, the studies in fields of city size distribution are limited [18]; therefore there is a dearth of research to help understand the effects of city size distribution and its environmental impacts. This is important as Ethiopia (the second most populous country in Africa) is known as the 'water tower' of Africa, having about 12 major river basins and 24 lakes (11 freshwater lakes, 9 saline lakes, and 4 crater lakes). Figure 1 shows that most of these lakes are located in the rift valley basin. Some are located in Awash basin (Koka, Gemari, and Abe), Central Rift Valley (CRV) basin (Ziway, Langano, Abijata, and Shala), and Southern basin (Hawassa, Abaya, Chamo, and Chew-Bahir) as the most important lakes [70].

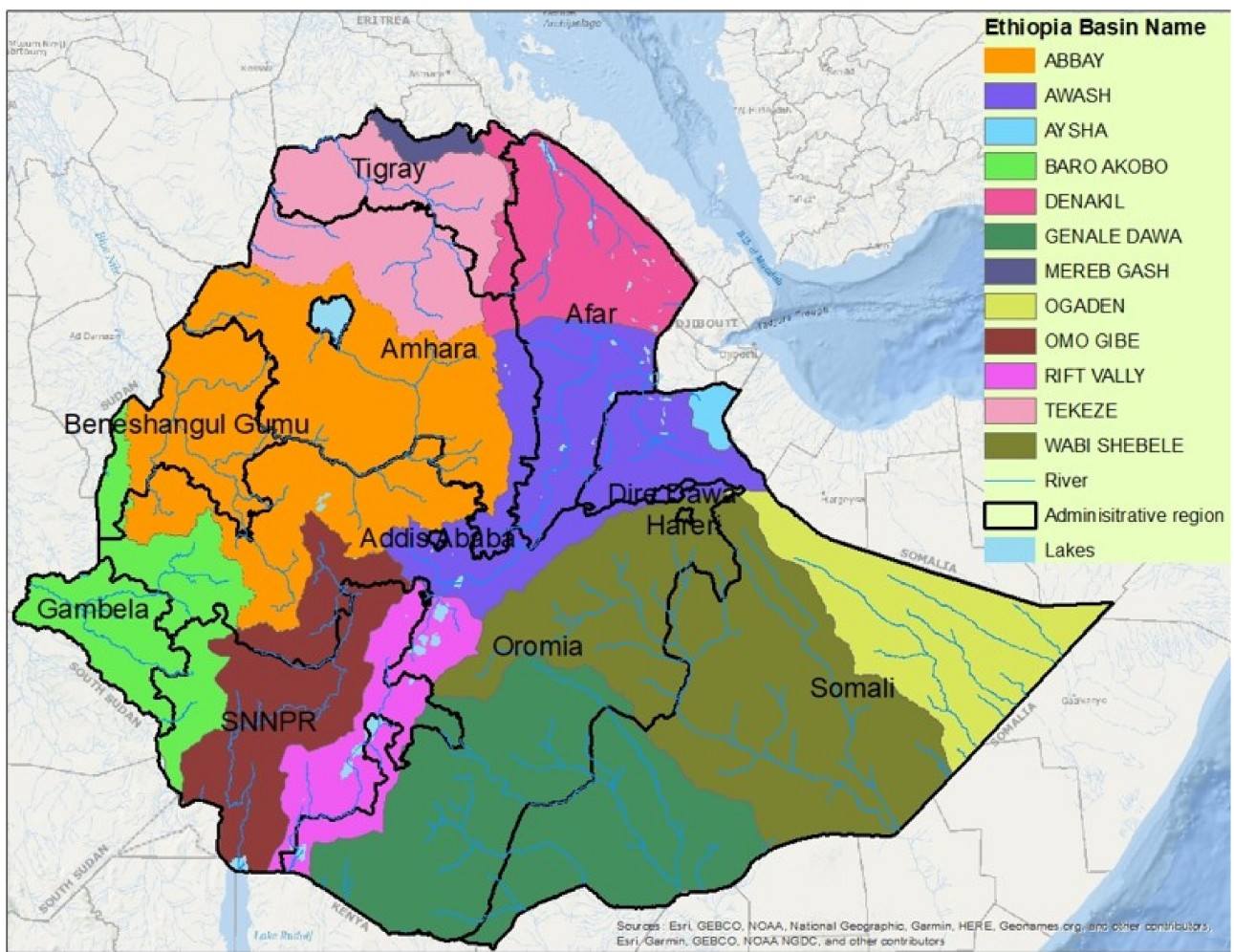

**Figure 1.** Map of Ethiopia: River basins, major rivers, and lakes. Source: Generated from shapefile and data from World Bank: available https://datacatalog.worldbank.org/search/dataset/0039833, accessed on 1 September 2022.

Except for four (Ziway, Tana, Langano, Abbaya and Chamo), most of the lakes are endorheic which means there is no surface water outlet. In other words, most of the lakes in Ethiopia are the end points of watersheds, which shows that the water generally stays within the boundaries of a lake's watershed [71]. Some of these lakes are located in close proximity to cities. The names of the cities and their corresponding lakes are Hawassa city—Lake Hawassa, Bahir Dar city—Lake Tana; Shashamane/Bishan Guracha—Lake Hawassa; Bishoftu—Lake Bishoftu; Arbaminch city—Lake Chamo and Abaya; Arsi Nagele city, Lake Langano; Ziway/Batu city—Lake Ziway; Meki town—Lake Ziway and Haramaya town, Lake Haramaya (Lake Haramaya, located 510 km east of Addis Ababa, is an extreme case which was once more than 10 miles around and 30 feet deep. currently it is no longer a lake).

In Ethiopia, the expansion of urbanization and industrialization around Ethiopia's Rift-Valley, where most of the lakes of the country are located, have exerted significant impacts on water quality and quantity [72,73]. The challenges that are attributed to urbanization include lack of sewerage and proper waste management systems which causes waste-water runoff and pollution to water resources. For instance, studies show that Lake Hawassa, one of the lakes located adjacent to Hawassa city, has been subject to several pollutants generated from industries such as textile, floury, sisal, soap and other factories, agriculture activities, service providing institutions such as hospitals that are located in and nearby the city, as well as the city urban storm water and sewerage discharged without treatment [73]. The effects of these industries, particularly the effluents from textile factory, sisal factory, soft drink factory, ceramic factory, and sewage, as well as the Hawassa referral hospital, makes the quality of the Lake Hawassa worse on the side of the city than the other side of the lake that shares a boundary with the rural districts [74,75].

### 2.2. Methods

In order to explore the relationship between the growth of urbanization and its implication for the management of CPRs (lakes), this study used various secondary data. It starts by reviewing various policy and planning documents, and rules and regulations directed towards urban development. Secondly, the study analyzes the geographical distribution of urban centers followed by the comparison of the growth of cities based on their proximity to lakes.

i.　**Understanding the geographical distribution of urban centers:** The Ministry of Urban Development and Construction (MUDCo) of Ethiopia classifies urban centers in terms of their population size into four groups, namely cities (>100,000), large towns (50,000–100,000), medium towns (20,000–50,000), and small towns (2000–20,000) (MUDCo, 2020). Following the classification of Ethiopia's urban centers, the population data of urban centers were collected from the CSA (census data of the years 1984, 1994, and 2007) and the MUDCo (2017). In the year 2017, the number of cities with a population size of 20,000 and above was 140 (the list of the 140 urban centers annexed (Appendix A)), which is about 94% of the population of the urban centers in the country. Hence, to explore the relationship between the lakes and urban centers, this study analyzes the geographical distribution of the 140 urban centers by using dot maps.

ii.　**Comparison between cities adjacent to lakes and those which are not:** To explore the relationship between urban centers and lakes in Ethiopia, we categorized the population of the 140 urban centers in the country into two groups, i.e., urban centers adjacent to lakes and urban centers not adjacent to lakes, and compared three periods: (1) between 1984 and 1994, (2) between 1994 and 2007, and (3) between 2007 and 2017. An independent sample *t*-test was conducted to test the rate of urbanization (rate of urbanization here refers to the percentage change observed in the population increase over the three periods for the two categories as well as the total number of urban population increase for the 140 urban centers) between these two groups of urban centers (i.e., cities adjacent to lakes and other cities and towns) and presented using tables.

The differences in the rates of urbanization between urban centers that are adjacent to lakes and various categories of urban centers were also examined using a one-way ANOVA test and post-hoc tests.

A one-way ANOVA requires one a categorical independent variable with three or more distinct categories and one continuous dependent variable; it shows whether there are significant differences in the mean scores of the dependent variable across more than two groups (e.g., across three groups) [76]. Hence, post-hoc tests can understand these differences. Further, the statistical significance of the difference between each pair of the groups was inferred from the post hoc test results.

To conduct a one-way between-groups analysis of variance and the post-hoc test, the 140 urban centers were divided into four groups: Group 1: urban centers adjacent to lakes (*n* = 9); Group 2: urban centers with a population of 100,000 and above (*n* = 16); Group 3:

urban centers with a population of 50,000–100,000 (*n* = 28); and Group 4: urban centers with a population of 20,000–50,000 (*n* = 87). These four groups were labelled as "lake cities and towns", "cities", "large towns", and "medium towns" respectively.

## 3. Results and Discussions

Ethiopia's urban planning proclamation defines a city as an 'urban center' with an 'established municipality or with a population size of 2000 or more inhabitants, in which 50 percent of the labor force is primarily engaged in non-agricultural activities' [77].

The Ethiopian urban system is characterized by two extremes: the primacy of Addis Ababa which hosts nearly a quarter of the urban population, and a large number of small towns below 20,000 inhabitants spread out thinly all over the country. However, looking into the number of cities by the urban population size, the overall trend in the urban hierarchy overtime reveals that Ethiopia has been experiencing a growing number of medium and large sized cities, and that these cities are expected to be potential urban centers with a significant role in undermining the dominance of the capital Addis Ababa. Figure 2 illustrates the structure of the urban system and trends in the hierarchy of cities over time.

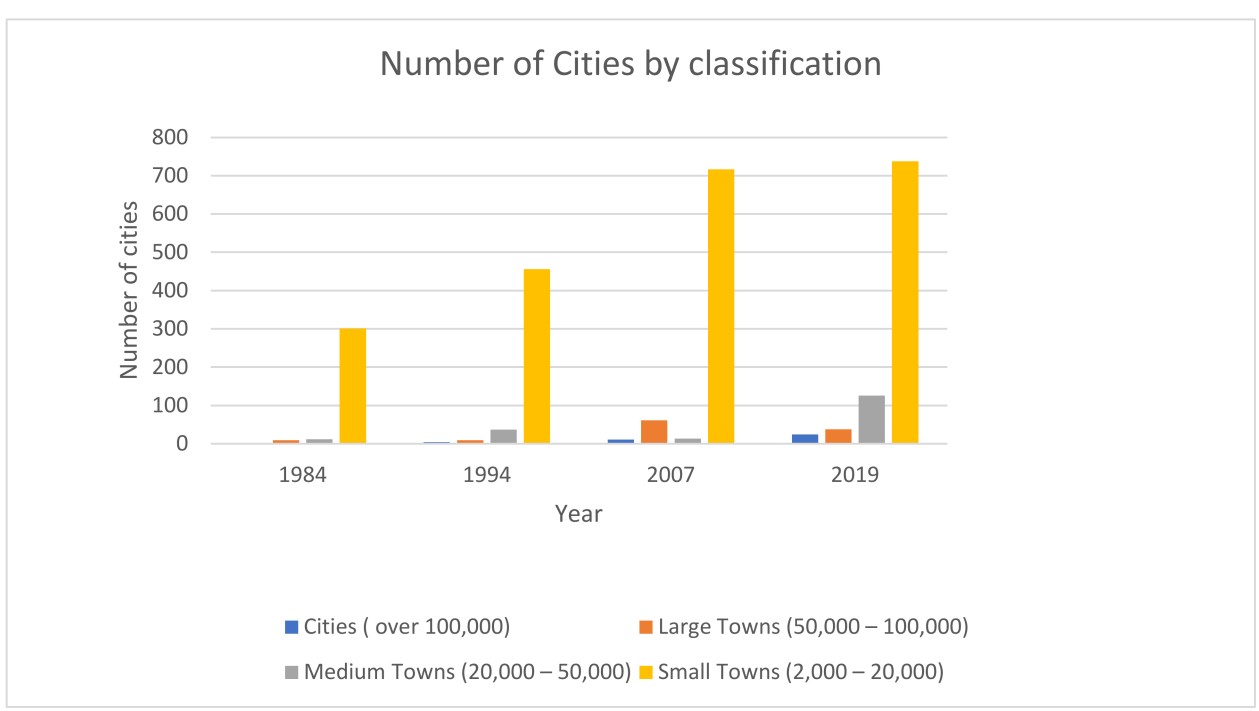

**Figure 2.** Number of cities by classification. Source: Computation based on CSA [78–81].

During the first census in Ethiopia, which was held in 1984, there were only 323 urban centers with over 2000 inhabitants in the country, out of which 301 were small towns with less than 20,000 inhabitants. By 2019, the total number of cities or urban centers had increased three-fold, reaching 926; however, it continued to be skewed towards smaller cities. In the same year, out of the 926 cities, only 23 towns exceeded a population size of 100,000 excluding Addis Ababa (i.e., Addis Ababa being a primate city with a population of over 3 million, whereas the majority of the urban centers (92%) have less than 20,000 inhabitants.

### 3.1. Spatial Distribution of Cities in Ethiopia

Notwithstanding the fact that the urban systems of Ethiopia follow transport corridors and are concentrated around Addis Ababa [82], the maps of the distribution of 140 urban centers in Figure 3 show that the concentration of cities around lakes is seen to have been significant over the last three decades.

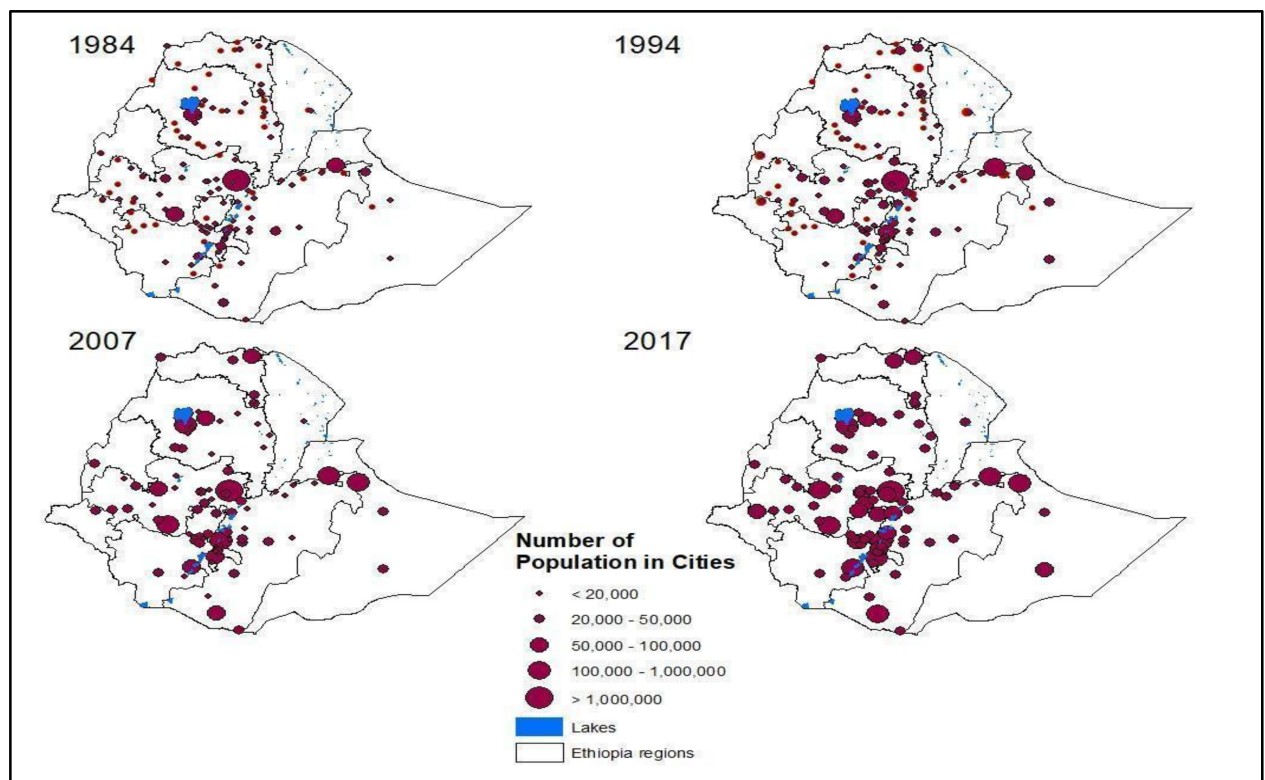

**Figure 3.** The major lakes and the distribution of cities in Ethiopia (1984–2017). Source: Illustrated by the author using data from CSA [78–80,83].

In the year 1984, due to the low level of urbanization, the concentration of cities in a specific corridor or location was seen to be not significant when compared with later decades. The concentration of the cities has increased over the last three decades. High concentration of cities in 2017 indicates that the growth rate of those cities that are located in close proximity to lakes is higher than in those cities without lakes.

### 3.2. Urban Centers and Lakes in Ethiopia

As presented in Section 3.1, the results of the distribution of cities in Ethiopia overlaid on the maps of lakes show that the concentration of cities has been significant around lakes. It is, therefore, necessary to analyze how the rate of urbanization of cities adjacent to lakes statistically differs from the rate of urbanization in other cities and towns in Ethiopia. Tables 1 and 2 present the descriptive statistics and the independent samples *t*-test results, respectively.

**Table 1.** Descriptive statistics on the rate of urbanization for cities adjacent to lakes and other cities and towns in Ethiopia over three periods.

| Periods/City Type | No. | First Period (1984–1994) % Change | | Second Period (1994–2007) % Change | | Third Period (2007–2017) % Change | |
|---|---|---|---|---|---|---|---|
| | | **Mean** | **St. dev** | **Mean** | **St. dev** | **Mean** | **St. dev** |
| **Cities adjacent to lakes** | 9 | 1.3868 | 0.80020 | 1.7396 | 1.06649 | 1.8480 | 1.54468 |
| **Other cities and towns** | 131 | 0.6681 | 0.71321 | 0.6438 | 0.79482 | 0.6364 | 0.62612 |
| **Total urban centers** | 140 | 0.7143 | 0.73748 | 0.7143 | 0.85384 | 0.7143 | 0.77001 |

Source: Computed by the author based on the population size figures for 1984, 1994, 2007, and 2017 [81].

**Table 2.** Independent sample *t*-test results on the rate of urbanization between cities adjacent to lakes and other urban areas over three periods.

| Period | Leven's Test for Equality of Variance | | *t*-Test for Equality of Means | |
|---|---|---|---|---|
| | F-Value | *p*-Value | *t*-Value | *p*-Value |
| First period (1984–1994) | 1.099 | 0.296 | 2.903 | 0.004 |
| Second period (1994–2007) | 2.851 | 0.094 | 3.911 | 0.000 |
| Third period (2007–2017) | 20.218 | 0.000 | 2.340 | 0.047 |

Source: Computed by the author based on the population size figures or data for 1984, 1994, 2007, and 2017 [81].

The increase in the total number of urban dwellers in 140 urban centers over ten years (from 1984 to 1994) was 1,408,906. Similarly, there was a total increase of 2,186,938 between 1994 and 2007 and a total increase of 3,854,484 urban dwellers in the third period between 2007 and 2017. The minimum and maximum percentage change observed per annum in the first period was −1.58% (decrease) for Wenji Gefersa town and 4.74% for Dire Dawa City. The average percentage increase for the 140 urban centers was 0.71% in the first period. The largest rate of urbanization was registered by the nine cities adjacent to lakes (i.e., an average of c. 1.4%) as compared with the average percentage change of c. 0.7% for the other 131 urban centers in the first period between 1984 and 1994.

The minimum and maximum percentage change observed per annum in the second period was −0.12% for Gode town and 5.44% for Mekele City. The largest rate of urbanization was attributed to the nine cities adjacent to lakes (i.e., an average of 1.74%) as compared with the average percentage change of 0.64% for the other 131 urban centers in the second period between 1994 and 2007.

The minimum and maximum percentage change observed per annum in the third period between 2007 and 2017 was 0.21% for three towns (i.e., Kofele, Huruta, and Deder) and 4.63% for Hawassa City, which is found adjacent to Lake Hawassa. The largest rate of urbanization was attributed to the nine cities adjacent to lakes (i.e., an average of 1.85%) as compared with the average percentage change of 0.64% for the other 131 urban centers in the third period. Overall, in this period, all the 140 urban centers have shown a positive change and even among the nine cities found adjacent to lakes, the highest urbanization rate was observed in Hawassa city.

In sum, there has been a continuous increase in the rate of urbanization for the cities adjacent to lakes between the three periods, whereas the percentage change was inconsistent for the other urban centers over the three periods. There was a slight decline in the observed maximum change for the later ones. The question here is whether such changes can be statistically confirmed. For this purpose, an independent sample *t*-test was conducted to test whether there is a statistical difference or not between the cities adjacent to lakes and the remaining urban centers in the proportional change in their urban population size over the three periods. Table 3 presents the results of the independent sample *t*-test.

As indicated in Table 3 there was a significant difference in the rate of urbanization between cities found adjacent to lakes and the other cities and towns in the first period [t (138) = 2.903, *p* = 0.004], second period [t (138) = 3.911, *p* = 0.000] and third period [t (138) = 2.340, *p* = 0.047]. That is, the cities adjacent to lakes have shown a significant difference from the other cities and towns in their rate of urbanization over the three periods or since 1984.

The above analysis shows that there is a significant difference between cities adjacent to lakes and the other cities and towns not adjacent to lakes. Nonetheless, there has been a faster urbanization rate in large cities such as Mekelle, and a declining or low urbanization rate in smaller towns. Hence, it is necessary to examine the difference in urbanization rates in terms of different categories of urban centers. Accordingly, the rate of urbanization in the nine cities found adjacent to lakes is compared against these three groups in the following section. Out of the nine cities; Hawassa, Bahirdar, Shashemene, Bishoftu, and Arbaminch

had a population of above 100,000 in 2017, whereas Arsi Negele, Ziway, Meki, and Haro Maya had a population between 50,000 and 100,000. These nine cities, which belong to either cities or large towns category, were recoded as "cities adjacent to lakes" in this study.

**Table 3.** Descriptive statistics of the four urban categories used in one way ANOVA to compare the rate of urbanization in the three periods.

| Periods | Urban Centre Categories | N | % | Annual Change of Urbanization Rate | | | |
|---|---|---|---|---|---|---|---|
| | | | | Min | Max | Mean | St. Dev |
| **First period (1984–1994)** | Lake cities and towns | 9 | 6.4 | 0.61 | 2.93 | 1.3868 | 0.80020 |
| | Cities | 16 | 11.4 | 0.67 | 4.74 | 1.7855 | 1.14770 |
| | Large towns | 28 | 20.0 | 0.24 | 3.25 | 0.7350 | 0.54916 |
| | Medium towns | 87 | 62.1 | −1.58 | 2.04 | 0.4411 | 0.39825 |
| | Total/average | 140 | 100.0 | −1.58 | 4.74 | 0.7143 | 0.73748 |
| **Second period (1994–2007)** | Lake cities and towns | 9 | 6.4 | 0.72 | 4.02 | 1.7396 | 1.06649 |
| | Cities | 16 | 11.4 | 0.60 | 5.44 | 2.0910 | 1.48653 |
| | Large towns | 28 | 20.0 | −0.12 | 1.78 | 0.7513 | 0.41027 |
| | Medium towns | 87 | 62.1 | 0.00 | 0.93 | 0.3431 | 0.17838 |
| | Total/average | 140 | 100.0 | −0.12 | 5.44 | 0.7143 | 0.85384 |
| **Third period (2007–2017)** | Lake cities and towns | 9 | 6.4 | 0.49 | 4.63 | 1.8480 | 1.54468 |
| | Cities | 16 | 11.4 | 0.98 | 3.98 | 1.9334 | 0.99050 |
| | Large towns | 28 | 20.0 | 0.34 | 1.28 | 0.7626 | 0.19372 |
| | Medium towns | 87 | 62.1 | 0.21 | 0.72 | 0.3573 | 0.11286 |
| | Total/average | 140 | 100.0 | 0.21 | 4.63 | 0.7143 | 0.77001 |

Source: Own computation (*n* = 140).

In a similar way to the above analysis, three premises are posited: (1) there is a significant difference between cities adjacent to lakes and other cities with 100,000 and above in their rate of urbanization in the three periods, (2) there is a significant difference between cities adjacent to lakes and large towns in their rate of urbanization in the three periods, and (3) there is a significant difference between cities adjacent to lakes and medium towns in their rate of urbanization in the three periods. Tables 3 and 4 present the descriptive statistics and the ANOVA test results in the rate of urbanization differences among these three groups of urban centers.

**Table 4.** ANOVA test results and test of homogeneity of variances of the four urban groups in terms of their difference in the rate of urbanization within three periods.

| Periods | Test of Homogeneity of Variances | | | | ANOVA | |
|---|---|---|---|---|---|---|
| | Leven Statistic | d.f1 | d.f2 | Sig. | F-Value | *p*-Value |
| **First period (1984–1994)** | 16.715 | 2 | 137 | 0.000 | 28.112 | 0.000 |
| **Second period (1994–2007)** | 58.158 | 2 | 137 | 0.000 | 47.423 | 0.000 |
| **Third period (2007–2017)** | 65.260 | 2 | 137 | 0.000 | 58.700 | 0.000 |

Source: Own computation (*n* = 140).

Table 3 above presents the descriptive statistics of four urban groups of the 140 urban centers. The first category comprises the nine cities and towns which are found adjacent to lakes and labeled as "lake cities and towns". The second, third, and fourth groups are categorized as "cities", "large towns", and "medium towns" per the MoUDC's classification as mentioned above, which comprise 11.49%, 20.0%, and 62.1% of the urban centers, respectively.

The differences in the rate of urbanization among the four groups of the urban centers were tested using a one-way ANOVA. As presented in Section 2.2, the ANOVA table provides evidence of whether there is a statistical difference among the mean scores of the

compared groups due to their difference in terms of the independent variables used which is provided in the column of significance values (*p*-values). In the current paper, this means the differences in the mean score of the four groups occur due to their differences in the urban category. The results of the one-way ANOVA analyses (F-value and *p*-value) and the Leven's test of homogeneity are presented under Table 4 below.

The results show that there was a statistically significant difference at the $p < 0.05$ in the mean scores of the four groups in terms of their rate of urbanization in the first period [F(3, 136) = 28.112, $p = 0.000$], in the second period [F(3, 136) = 47.423, $p = 0.000$] and third period [F(3, 136) = 58.700, $p = 0.000$]. As compared with the third and fourth groups, the rate of urbanization in the first group is significantly different in all three periods. Similarly, a statistically significant difference was observed between group two and group three in all three periods. There was a statistically significant difference between group three and group four in the second and third periods, but not in the first period. There was no statistical difference in the rate of urbanization between group one and group two in all three periods. In other words, the "lake cities and towns" showed a statistically significant difference in their rate of urbanization as compared with the "large towns" and "medium towns" groups in the three periods, but not with the "cities" group.

The one-way ANOVA test makes a number of assumptions which are similar to the underlying assumptions considered for the independent samples *t*-test above. Likewise, the data set of this study ($n = 140$) does not violate the underlying assumptions in conducting a one-way ANOVA test except for test of homogeneity of variances. As can be seen from Table 4 above, the Leven's test for homogeneity of variance assumption, which tests whether the variance in scores for each of the four groups is less than 0.05, was violated [76]. According to [76], the violation of this assumption does not pose a serious problem for the robustness of the results in the ANOVA test, as long as the ANOVA results are significant (i.e., $p < 0.05$). Once the suitability of the data was proved for the use of ANOVA test, the main premises of this study, which state that the rate of urbanization among cities and towns found adjacent to lakes is significantly different from the rate of urbanization for the other three urban center groups, were found to be true, since the *p*-values are significant ($p = 0.000$) in all of the three periods, as depicted in Table 4. Therefore, we expect a statistical difference among the four urban center groups in terms of their rate of urbanization in the three periods. Nonetheless, these significant ANOVA results do not reveal which group is different from the other groups.

The statistical significance of the difference between each pair of groups can be inferred from the post-hoc test results [76]. Therefore, after obtaining a statistically significant difference, post hoc tests were used to tell the difference in the rate of urbanization among the four compared groups of urban centers in the three periods. Hence, in this section, the post-hoc test results where significant differences are expected among each of the pairs compared are presented and discussed in detail. In specific terms, the results of the one-way between-groups analysis of variance with post-hoc tests are presented and discussed for the four groups being compared, which are significantly different from one another below $p < 0.05$ (see Table 5).

Post-hoc comparisons using the Tukey HSD test indicated that the rate of urbanization observed between the lake cities and towns was significantly different from the rate of urbanization among the large towns in the first period ($p = 0.022$), second period ($p = 0.000$), and third period ($p = 0.000$). From the positive results in mean score differences between these two groups in all the three periods, it can be inferred that the lake cities and towns group had experienced a much higher urbanization rate, which was more pronounced during the third period since the mean difference was 1.085%. Similarly, the mean score difference between the lake cities and towns group and medium towns group was significant in the first period ($p = 0.000$), second period ($p = 0.000$), and third period ($p = 0.000$). From the positive mean difference results, it can be inferred that the urban centers adjacent to lakes had experienced a higher rate of urbanization in all the three periods as compared with the medium towns and this difference was more pronounced in the third period.

**Table 5.** Post-hoc test results showing multiple comparisons for the rate of urbanization differences (*p*-values) among the four urban center categories in the three periods.

| Urban Categories | Multiple Comparison | Periods | | | | | |
|---|---|---|---|---|---|---|---|
| | | First Period (1984–1994) | | Second Period (1994–2007) | | Third Period (2007–2017) | |
| | | Mean Difference | *p*-Value | Mean Difference | *p*-Value | Mean Difference | *p*-Value |
| **Lake cities & towns** | Cities | −0.39869 | 0.363 | −0.35142 | 0.503 | −0.08533 | 0.978 |
| | Large towns | 0.65177 * | 0.022 | 0.98827 * | 0.000 | 1.08549 * | 0.000 |
| | Medium towns | 0.94572 * | 0.000 | 1.39646 * | 0.000 | 1.49079 * | 0.000 |
| **Cities** | Lake cities and towns | 0.39869 | 0.363 | 0.35142 | 0.503 | 0.08533 | 0.978 |
| | Large towns | 1.05047 * | 0.000 | 1.33969 * | 0.000 | 1.17082 * | 0.000 |
| | Medium towns | 1.34442 * | 0.000 | 1.74787 * | 0.000 | 1.57612 * | 0.000 |
| **Large towns** | Lake cities & towns | −0.65177 * | 0.022 | −0.98827 * | 0.000 | −1.08549 * | 0.000 |
| | Cities | −1.05047 * | 0.000 | −1.33969 * | 0.000 | −1.17082 * | 0.000 |
| | Medium towns | 0.29395 | 0.101 | 0.40818 * | 0.012 | 0.40530 * | 0.002 |
| **Medium towns** | Lake cities and towns | −0.94572 * | 0.000 | −1.39646 * | 0.000 | −1.49079 * | 0.000 |
| | Cities | −1.34442 * | 0.000 | −1.74787 * | 0.000 | −1.57612 * | 0.000 |
| | Large towns | −0.29395 | 0.101 | −0.40818 * | 0.012 | −0.40530 * | 0.002 |

Source: Own computations (*n* = 140). * the differences in the mean values are statistically significant.

In contrast, post-hoc comparisons using the Tukey HSD test indicated that the rate of urbanization observed among the lake cities and towns was not significantly different from the rate of urbanization observed among the cities group with a population size above 100,000 in the first period (*p* = 0.363), second period (*p* = 0.503), and third period (*p* = 0.978). This shows there is no statistical support to claim that the urban centers adjacent to lakes had experienced a higher rate of urbanization than cities in all the three periods. This result may be attributed to the administrative functions of the urban centers grouped under "cities". Cities such as Mekelle, Harar, and Jigjiga have been serving as regional capitals since 1995, while cities such as Gondar, Dessie, Jimma, Nekemte, Assela, and Debremarkos had been among the 14 administrative capitals prior to 1991. Moreover, in some cities such as Adama, which was not serving as higher administrative center before 1995, other factors such as commercial activities and location as junctures with higher road and railway connectivity might have contributed to a higher rate of urbanization. Despite its being a chartered city under the federal government since 1995, Dire Dawa has experienced a similar situation to Adama in terms of railway connectivity and commercial functions. Commercial and industrial functions among cities such as Kombolcha, Hossana, Debrebirhan, Debretabor, and Dila might have also contributed to a higher rate of urbanization among the urban centers in the "cities" group.

In a nutshell, lakes are found to be driving factors of a higher rate of urbanization in an Ethiopian context. This is an interesting finding since Ethiopia is a land-locked country which cannot experience an emergence of urban centers along coastal lines or a faster growing rate in such geographies. Nonetheless, this does not mean that lakes are the leading factor of higher rates of urbanization among Ethiopian cities and towns. Other factors such as administrative and commercial and transport connectivity may also contribute to it. However, the latter needs a further investigation, which will be an avenue for future research.

## 4. Conclusions and Recommendations

The aim of this study was to assess the relationship between the expansion of urban centers and lakes as common pool resources in the Ethiopian context, and thereby to draw conclusions about the implication of such relationships, followed by policy recommendations.

Ethiopia's urban centers were thought to be concentrated around highlands, far away from main water bodies such as lakes and rivers. The findings of this study reveal the emergence of changes, or a shift, in the process of urbanization in the country with the expansion of cities towards water bodies or lakes.

By assessing the distribution of 140 cities (with a population over 20,000) in Ethiopia, it is found that, over the last three decades (1984–2017), urban areas located near or adjacent to lakes have experienced a faster growth rate than the urban centers located far away from lakes. The difference between the two categories, (i) nine cities adjacent to lakes and (ii) 131 other cities—not adjacent, was found to be statistically significant. Based on the faster growth rate of urbanization around lakes in Ethiopia, this study argues that lakes are fostering factors for higher rates of urbanization in an Ethiopian context; however, there exist several other factors that may contribute such as policies, administrative factors, and transport connectivity, which needs further investigation.

Lakes as common pool resources are more vulnerable to urban and human activities than other natural resources [54]. The impacts of urban and human activities include the decrease in the lake area and the influence of pollution on water quality [14,55]. The role of cities in affecting the large scale ecosystems within their surroundings is significantly high, making it critical to understand how changes in urban land use and governance affect the use of urban ecosystems [56]. The significance of such studies is more important in the developing world as the changes in urban land which are caused by urbanization are rapid and unplanned [57]. Rapid urbanization around lakes in Ethiopia has been considered a threat to the quality and volume of lakes. The expansion of cities in the country is mostly unplanned with no buffer zones and proper waste management systems which exacerbates the vulnerability of lakes that are located nearby. Hence, this study argues that the management of lakes in Ethiopia relies on how cities are planned and managed.

Moreover, this study not only explores a shift from rural to urban-dominated CPR systems (i.e., rapid urban growth around lakes) but also serves as an addition to the limited studies that link urbanization with CPRs as noted by scholars in the field [47,57,84,85]. Considering the changing situation around lakes (i.e., the rapid expansion of urban centers around lakes, as argued by this study), we recommend that future research of CPRs should further analyze the effect of urbanization as well as the relationship between lakes and urban centers while undertaking research concerning CPR management. This study, therefore, serves as a springboard for researchers to conduct similar studies by considering lakes as CPRs in a context of urbanization. Given the rapid growth of urban centers around the lakes in Ethiopia, policies, rules, and regulations that are directed towards urbanization play a key role in determining the current and future states of Ethiopian lakes.

**Author Contributions:** Supervision, E.O., F.B. and W.A.; Writing—original draft, A.F.K. All authors have read and agreed to the published version of the manuscript.

**Funding:** This research received no external funding.

**Institutional Review Board Statement:** Not applicable.

**Informed Consent Statement:** Not applicable.

**Conflicts of Interest:** The authors declare no conflict of interest.

## Appendix A

**Table A1.** Cities and Towns of Ethiopia with 20,000 population and more during 1984, 1994, and 2007 Censuses, and Projections from 2013 to 2017.

| Ser. No. | Name | Region or Administration | Population Census | | | CSA Projection 2017 |
|---|---|---|---|---|---|---|
| | | | 1984 | 1994 | 2007 | 2017 |
| | 1 | 2 | 3 | 4 | 5 | 6 |
| 1 | Addis Ababa | Addis Ababa | 1,412,575 | 2,112,737 | 2,739,551 | 3,434,000 |
| 2 | Mekele | Tigray | 61,583 | 96,938 | 215,914 | 358,529 |
| 3 | Adama | Oromia | 76,284 | 127,842 | 220,212 | 355,475 |
| 4 | Dire Dawa | Dire Dawa | 98,104 | 164,851 | 233,224 | 293,000 |
| 5 | Gondar | Amhara | 80,886 | 112,249 | 207,044 | 360,600 |
| 6 | Hawassa | SNNPR | 36,169 | 69,169 | 157,139 | 335,508 |
| 7 | Bahir Dar | Amhara | 54,800 | 96,140 | 155,428 | 313,997 |
| 8 | Jimma | Oromia | 60,992 | 88,867 | 120,960 | 195,228 |
| 9 | Dessie | Amhara | 68,848 | 97,314 | 120,095 | 209,226 |
| 10 | Jijiga | Somaliya | 23,183 | 56,821 | 125,876 | 169,390 |
| 11 | Shashamane | Oromia | 31,531 | 52,080 | 100,454 | 162,127 |
| 12 | Bishoftu (Debre Zeyit) | Oromia | 51,143 | 73,372 | 99,928 | 161,354 |
| 13 | Harar | Harari | 62,160 | 76,378 | 99,368 | 137,000 |
| 14 | Sodo | SNNPR | 24,592 | 36,287 | 76,050 | 161,450 |
| 15 | Arba Minch | SNNPR | 23,032 | 40,020 | 74,879 | 159,019 |
| 16 | Hosaina | SNNPR | 15,167 | 31,701 | 69,995 | 148,847 |
| 17 | Nekemte | Oromia | 28,824 | 47,258 | 75,219 | 121,385 |
| 18 | Asella | Oromia | 36,720 | 47,391 | 67,269 | 108,571 |
| 19 | Dila | SNNPR | 23,936 | 33,734 | 59,150 | 125,599 |
| 20 | Debre Birhan [Debre Berhan] | Amhara | 25,753 | 38,717 | 65,231 | 113,693 |
| 21 | Debre Markos [Debre Marqos] | Amhara | 39,808 | 49,297 | 62,497 | 108,882 |
| 22 | Adigrat | Tigray | 16,262 | 37,417 | 57,588 | 95,358 |
| 23 | Kombolcha | Amhara | 15,782 | 39,466 | 58,667 | 102,244 |
| 24 | Debre Tabor | Amhara | 15,306 | 22,455 | 55,596 | 96,973 |
| 25 | Gambela | Gambella | 4492 | 18,263 | 39,022 | 74,102 |
| 26 | Sebeta | Oromia | 10,030 | 14,076 | 49,331 | 79,633 |
| 27 | Burayu | Oromia | . . . | 10,027 | 48,876 | 78,902 |
| 28 | Enda Silase (Shire-Enda Silase) | Tigray | 12,846 | 25,269 | 47,284 | 78,366 |
| 29 | Ambo | Oromia | 17,325 | 27,636 | 48,171 | 77,735 |
| 30 | Arsi Negele | Oromia | 13,096 | 23,512 | 47,292 | 76,340 |
| 31 | Aksum [Axum] | Tigray | 17,753 | 27,148 | 44,647 | 74,007 |
| 32 | Woldiya | Amhara | 15,690 | 24,533 | 46,139 | 80,484 |
| 33 | Robe (Bale Zone) | Oromia | 11,293 | 21,516 | 44,382 | 71,625 |
| 34 | Ziway (Batu) | Oromia | 6585 | 20,056 | 43,660 | 70,436 |
| 35 | Adwa | Tigray | 13,823 | 24,519 | 40,500 | 67,065 |
| 36 | Gode | Somaliya | . . . | 45,755 | 43,234 | 56,398 |
| 37 | Woliso | Oromia | 16,811 | 25,491 | 37,878 | 61,140 |
| 38 | Butajira | SNNPR | 13,688 | 20,509 | 33,406 | 71,045 |
| 39 | Meki | Oromia | 11,168 | 20,460 | 36,252 | 58,490 |
| 40 | Negele [Negele Boran] | Oromia | 11,997 | 23,997 | 35,264 | 56,897 |
| 41 | Areka | SNNPR | 4231 | 12,294 | 31,408 | 66,815 |
| 42 | Alamata | Tigray | 14,030 | 26,179 | 33,214 | 55,153 |
| 43 | Yirga Alem | SNNPR | 16,003 | 24,183 | 30,348 | 64,507 |

**Table A1.** *Cont.*

| Ser. No. | Name | Region or Administration | Population Census | | | CSA Projection 2017 |
|---|---|---|---|---|---|---|
| | | | 1984 | 1994 | 2007 | 2017 |
| | 1 | 2 | 3 | 4 | 5 | 6 |
| 44 | Chiro (Asebe Teferi) | Oromia | 11,344 | 18,678 | 33,670 | 54,307 |
| 45 | Welkite | SNNPR | 7855 | 15,329 | 28,866 | 61,309 |
| 46 | Goba | Oromia | 22,963 | 28,358 | 32,025 | 51,715 |
| 47 | Asosa | Benishangul Gumuz | 4159 | 11,749 | 24,214 | 52,575 |
| 48 | Wukro | Tigray | 13,045 | 16,421 | 30,210 | 50,080 |
| 49 | Gimbi | Oromia | 13,098 | 20,462 | 30,981 | 49,999 |
| 50 | Haro Maya | Oromia | . . . | 8560 | 30,728 | 49,584 |
| 51 | Alaba Kulito | SNNPR | 8902 | 15,101 | 26,867 | 57,076 |
| 52 | Mojo | Oromia | 13,945 | 21,997 | 29,547 | 47,704 |
| 53 | Dembi Dolo | Oromia | 14,170 | 19,587 | 29,448 | 47,519 |
| 54 | Metu | Oromia | 12,491 | 19,298 | 28,782 | 46,456 |
| 55 | Degehabur | Somaliya | . . . | 28,708 | 30,027 | 40,386 |
| 56 | Moyale | Oromia | 4038 | 10,543 | 28,056 | 44,459 |
| 57 | Bule Hora (Hagere Mariam) | Oromia | 7327 | 12,718 | 27,820 | 44,885 |
| 58 | Tepi | SNNPR | 4459 | 10,616 | 24,829 | 52,719 |
| 59 | Kebri Dahar | Somaliya | . . . | 24,263 | 29,241 | 39,315 |
| 60 | Fiche | Oromia | 17,106 | 21,187 | 27,493 | 44,400 |
| 61 | Durame | SNNPR | . . . | 7092 | 24,472 | 52,084 |
| 62 | Boditi | SNNPR | 4403 | 13,400 | 24,133 | 51,324 |
| 63 | Mota (Hulet Ej Enese) | Amhara | 12,934 | 18,160 | 26,177 | 45,693 |
| 64 | Finote Selam | Amhara | 8156 | 13,834 | 25,913 | 45,215 |
| 65 | Mizan Teferi (Mizan-Aman) | SNNPR | . . . | 10,652 | 23,144 | 72,324 |
| 66 | Agaro | Oromia | 18,764 | 23,246 | 25,458 | 41,085 |
| 67 | Sawla (Felege Neway) | SNNPR | 7526 | 15,764 | 22,704 | 48,277 |
| 68 | Dolo | Somaliya | . . . | 20,762 | 26,232 | 35,398 |
| 69 | Dangila | Amhara | 10,602 | 15,437 | 24,827 | 43,308 |
| 70 | Kobo | Amhara | 13,542 | 20,788 | 24,867 | 43,376 |
| 71 | Aleta Wendo | SNNPR | 9685 | 11,321 | 22,093 | 46,905 |
| 72 | Maychew | Tigray | 14,190 | 19,757 | 23,419 | 38,839 |
| 73 | Bonga | SNNPR | . . . | 10,851 | 20,858 | 44,329 |
| 74 | Holeta | Oromia | 11,741 | 16,785 | 23,296 | 37,606 |
| 75 | Chagni | Amhara | 8421 | 17,777 | 23,232 | 40,498 |
| 76 | Adola (Kebre Mengist) | Oromia | 14,391 | 20,136 | 22,938 | 37,016 |
| 77 | Shakiso | Oromia | 7032 | 15,757 | 22,930 | 36,990 |
| 78 | Jinka | SNNPR | 4480 | 12,407 | 20,267 | 43,020 |
| 79 | Humera | Tigray | 10,469 | 14,451 | 21,653 | 36,074 |
| 80 | Sekota (Soqota) | Amhara | . . . | 7922 | 22,346 | 38,937 |
| 81 | Werota (Wereta) | Amhara | 8614 | 15,181 | 21,222 | 37,011 |
| 82 | Injibara (Banja Shekudaa Woreda) | Amhara | . . . | 754 | 21,065 | 36,757 |
| 83 | Dodola | Oromia | 8287 | 13,847 | 20,830 | 33,605 |
| 84 | Debark′ [Debarq] | Amhara | 8484 | 14,474 | 20,839 | 36,244 |
| 85 | Asasa | Oromia | 5068 | 10,903 | 20,667 | 33,354 |
| 86 | Chuko | SNNPR | . . . | 4583 | 18,467 | 39,171 |

**Table A1.** *Cont.*

| Ser. No. | Name | Region or Administration | Population Census | | | CSA Projection 2017 |
|---|---|---|---|---|---|---|
| | | | 1984 | 1994 | 2007 | 2017 |
| | 1 | 2 | 3 | 4 | 5 | 6 |
| 87 | Bure | Amhara | 8177 | 13,437 | 20,410 | 35,622 |
| 88 | Hadero | SNNPR | . . . | 4482 | 17,831 | 37,933 |
| 89 | Gebre Guracha (Kuyu) | Oromia | 7394 | 11,113 | 19,872 | 32,076 |
| 90 | Nefas Mewcha | Amhara | 6548 | 10,808 | 19,620 | 34,195 |
| 91 | Bedele | Oromia | 6988 | 11,907 | 19,517 | 31,500 |
| 92 | Kemise | Amhara | 4721 | 10,822 | 19,420 | 33,887 |
| 93 | Adet | Amhara | 6501 | 12,178 | 19,169 | 33,445 |
| 94 | Nejo | Oromia | 6160 | 11,125 | 18,998 | 30,657 |
| 95 | Asayita | Affar | . . . | 15,475 | 16,052 | 29,963 |
| 96 | Mer Awi | Amhara | . . . | 9282 | 18,682 | 32,624 |
| 97 | Bedessa | Oromia | 6654 | 10,813 | 18,187 | 29,340 |
| 98 | Ginchi | Oromia | 6487 | 10,592 | 18,134 | 29,262 |
| 99 | Babille | Oromia | . . . | 9195 | 17,712 | 28,590 |
| 100 | Bekoji | Oromia | . . . | 9367 | 17,741 | 28,635 |
| 101 | Shiraro | Tigray | . . . | 8415 | 17,045 | 28,287 |
| 102 | Awash Sebat Kilo | Affar | 8684 | . . . | 14,880 | 27,759 |
| 103 | Shewa Robit (Kewet) | Amhara | 9783 | 14,287 | 17,575 | 30,670 |
| 104 | Yabelo | Oromia | 5985 | 10,322 | 17,497 | 28,222 |
| 105 | Shone | SNNPR | . . . | 8230 | 15,616 | 33,174 |
| 106 | Korem | Tigray | 9348 | 16,895 | 16,856 | 27,900 |
| 107 | Dubti | Affar | . . . | . . . | 14,715 | 27,474 |
| 108 | Lalibela | Amhara | . . . | 8484 | 17,367 | 30,235 |
| 109 | Tis Abay | Amhara | . . . | 4227 | 17,370 | 30,319 |
| 110 | Ginir | Oromia | 8594 | 12,068 | 17,102 | 27,598 |
| 111 | Yirga Chefe | SNNPR | 8291 | 11,579 | 15,118 | 32,134 |
| 112 | Bati | Amhara | 10,009 | 13,965 | 16,710 | 29,084 |
| 113 | Abiy Addi | Tigray | . . . | 7884 | 16,115 | 26,759 |
| 114 | Logia | Affar | . . . | . . . | 14,038 | 26,230 |
| 115 | Gelemso | Oromia | 7271 | 10,849 | 16,484 | 26,584 |
| 116 | Bako | Oromia | 6081 | 10,422 | 16,445 | 26,530 |
| 117 | Bichena | Amhara | . . . | 12,484 | 16,206 | 28,266 |
| 118 | Adis Zemen | Amhara | 9093 | 14,342 | 16,113 | 28,122 |
| 119 | Mersa | Amhara | . . . | 7274 | 16,122 | 28,123 |
| 120 | Shinshicho | SNNPR | . . . | 6968 | 14,285 | 33,714 |
| 121 | Robe (Arsi Zone) | Oromia | . . . | 9599 | 15,169 | 24,468 |
| 122 | Welenchiti (Boset Woreda) | Oromia | 7419 | 11,732 | 15,183 | 24,508 |
| 123 | Ayikel (Chilga) | Amhara | . . . | 8364 | 15,127 | 26,316 |
| 124 | Eteya (Hitosa Woreda) | Oromia | . . . | 7260 | 14,985 | 24,192 |
| 125 | Shambu | Oromia | 8252 | 11,327 | 14,995 | 24,196 |
| 126 | Dera | Oromia | . . . | 9356 | 14,786 | 23,869 |
| 127 | Guder | Oromia | . . . | 9562 | 14,742 | 23,799 |
| 128 | Gidole | SNNPR | . . . | 8167 | 13,184 | 28,070 |
| 129 | Abomsa | Oromia | 7489 | 10,742 | 14,655 | 23,637 |
| 130 | Tulu Bolo | Oromia | . . . | 8011 | 14,476 | 23,366 |
| 131 | Mehoni (Raya Azebo) | Tigray | . . . | . . . | 13,793 | 22,885 |
| 132 | Wenji Gefersa | Oromia | 35,420 | 13,156 | 14,060 | 22,702 |

**Table A1.** *Cont.*

| Ser. No. | Name | Region or Administration | Population Census | | | CSA Projection 2017 |
|---|---|---|---|---|---|---|
| | | | 1984 | 1994 | 2007 | 2017 |
| | 1 | 2 | 3 | 4 | 5 | 6 |
| 133 | Mendi (Menesibu Woreda) | Oromia | 3778 | 10,070 | 14,008 | 22,608 |
| 134 | Este (Misrak Este) | Amhara | . . . | 9241 | 13,901 | 24,258 |
| 135 | Gutin | Oromia | . . . | 2770 | 13,641 | 22,013 |
| 136 | Kofele | Oromia | . . . | 7336 | 13,483 | 21,747 |
| 137 | Mieso | Oromia | . . . | 5769 | 13,339 | 40,972 |
| 138 | May Cadera (Humera Woreda) | Tigray | . . . | . . . | 12,850 | 21,393 |
| 139 | Huruta (Lude Hitosa-Woreda) | Oromia | . . . | 9465 | 13,265 | 21,414 |
| 140 | Leku | SNNPR | . . . | 8671 | 11,831 | 25,107 |
| 141 | Deder | Oromia | . . . | 6758 | 12,967 | 20,916 |

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
