# Peer review of "Rapid Urbanization in Ethiopia: Lakes as Drivers and Its Implication for the Management of Common Pool Resources"

_sustainability, doi:10.3390/su141912788_

Round 1

Reviewer 1 Report

1- The abstract should be a single paragraph.

2- No need to have abbreviations in keywords.

3- The introduction section is too long with four sub-sections. This style is not academic. I recommend the authors shorten the introduction section and then move those subsections to a separate section as the theoretical background or literature review section.

4- In the method section the authors have written that “In order to explore the relationship between the growth of urbanization and its im-255 plication to the management of CPRs (lakes), this study used various secondary data”. But there is no clear information about the data. Please provide better clarification for this, including what type of data? Where? Is there any link to access them?

5- The quality of Figure 2 is not good.

Author Response

The abstract should be a single paragraph.

The abstract changed into a single paragraph (see changes in red inks)

No need to have abbreviations in keywords.

For Keywords abbreviation - (CPRs) deleted

3- The introduction section is too long with four sub-sections. This style is not academic. I recommend the authors shorten the introduction section and then move those subsections to a separate section as the theoretical background or literature review section.

The introduction section modified (started with the second section and moved the last two sections to the methods and material sections – by cutting the paragraph that narrates abut Africa

4- In the method section the authors have written that “In order to explore the relationship between the growth of urbanization and its im-255 plication to the management of CPRs (lakes), this study used various secondary data”. But there is no clear information about the data. Please provide better clarification for this, including what type of data? Where? Is there any link to access them?

Secondary data include: i) Ethiopia’s urban population data – collected from census reports and unpublished report of the ministry of urban development and Construction) - Urban population data annexed as part of this article

  1. ii) policy and strategy documents review (some of the documents reviewed are:
  • Ethiopian urban development policy (The policy is available (Amharic version available in hard copies)
  • Urban planning strategy or planning regulations (There are about eight strategy documents (most of which are Amharic (are available in hard copies)
  • Urban planning regulation/manual

Moreover, number (i) and (2) of the methodology section provide clarifications

5- The quality of Figure 2 is not good.

Improved (substituted by better quality figure)

Reviewer 2 Report

The paper is in the journal’s scope, and to my opinion, a valuable piece for the special issue ‘Sustainable Urban and Rural Development’.

The study offers elements of both conceptual advance and analysis to improve the understanding around the topic, and in terms of case-study novelty. The issue is an overlooked one, to my opinion, and also as the literature suggests:

Although the design, similar findings, and the general approach have been reported in the literature, there are only a few publications focusing on Africa, and Ethiopia, with the exception of the paper by Fitawok et al. (2020).

The analysis seems thorough, with a big dataset and time-span.

·         I recommend making the datasets used available, as this would be a major added value from your work. These study areas suffer from data limitations.

Although the main message is not surprising and expected, that lakes are fostering factors for a higher rate of urbanization, this study documents this by providing evidence, useful to raise awareness for a more reasonable management.

·         I recommend further stressing and elaborating on this aspect, by considering the environmental and economic implications of managing limited resources under competitive needs (and under climate variability). Check also these references that might be helpful:

a) Loucks, D.P. Managing Water as a Critical Component of a Changing World. Water Resour Manage 31, 2905–2916 (2017). https://doi.org/10.1007/s11269-017-1705-7 

b) Alamanos A (2021) Sustainable water resources management under water-scarce and limited-data conditions. Central Asian Journal of Water Research 7:1–19. https://doi.org/10.29258/CAJWR/2021-R1.v7-2/1-19.eng

Minor comments:

1.       The references, especially in the introduction, seem to be quite old, often more than a decade ago. Since these fields have been evolving, I recommend adding some recent relevant references. See examples below, hopefully helpful:

·         Fitawok MB, Derudder B, Minale AS, et al (2020) Modeling the Impact of Urbanization on Land-Use Change in Bahir Dar City, Ethiopia: An Integrated Cellular Automata–Markov Chain Approach. Land 9:115. https://doi.org/10.3390/land9040115

·         Lynch LS (2016) Beyond The Greenbelt: Extended Urbanization On The Shores Of Lake Simcoe

·         Rajendran K (2021) Conservation and Protection of Peri-Urban Rural Landscapes from the Impacts of Urbanization: Case Study of Manimangalam, Mahanyam and Malaipattu Villages in Manimangalam Watershed. In: Thirumaran K, Balaji G, Prasad ND (eds) Sustainable Urban Architecture. Springer, Singapore, pp 17–34

·         Englezos N., Kartala X., Koundouri P., Tsionas M. & Alamanos, A. (2021). A novel hydro-economic – econometric approach for integrated transboundary water management under uncertainty. DEOS Working Papers. http://wpa.deos.aueb.gr/docs/INTEGRATED.TRANSBOUNDARY.WATER.MANAGEMENT.pdf

·         Wang T, Wang L, Ning Z-Z (2020) Spatial pattern of tourist attractions and its influencing factors in China. Journal of Spatial Science 65:327–344. https://doi.org/10.1080/14498596.2018.1494058

2.       The reference style does not seem the usual one of mdpi with numbering.

Author Response

Comments and Suggestions for Authors

I recommend making the datasets used available, as this would be a major added value from your work. These study areas suffer from data limitations.

Urban population data annexed to this article

Although the main message is not surprising and expected, that lakes are fostering factors for a higher rate of urbanization, this study documents this by providing evidence, useful to raise awareness for a more reasonable management.

  • I recommend further stressing and elaborating on this aspect, by considering the environmental and economic implications of managing limited resources under competitive needs (and under climate variability). Check also these references that might be helpful:

As it requires more data and expansion of the scope, I am working on additional study/article/ on the implication of such relationship between lakes and urban centers (which is also believed to be an addition to the studies of the field)

a) Loucks, D.P. Managing Water as a Critical Component of a Changing World. Water Resour Manage31, 2905–2916 (2017). https://doi.org/10.1007/s11269-017-1705-7 

b) Alamanos A (2021) Sustainable water resources management under water-scarce and limited-data conditions. Central Asian Journal of Water Research 7:1–https://doi.org/10.29258/CAJWR/2021-R1.v7-2/1-19.eng

 Minor comments:

  1. The references, especially in the introduction, seem to be quite old, often more than a decade ago. Since these fields have been evolving, I recommend adding some recent relevant references. See examples below, hopefully helpful:

Fitawok et al. 2020, cited. With regards to some of the suggested references are similar concussions of already sited/refferd artcles such as  Nagendra and Ostrom 2014, Wang et al., 2020, Yang et al., 2020

  • Fitawok MB, Derudder B, Minale AS, et al (2020) Modeling the Impact of Urbanization on Land-Use Change in Bahir Dar City, Ethiopia: An Integrated Cellular Automata–Markov Chain Approach. Land 9:115. https://doi.org/10.3390/land9040115

  • Lynch LS (2016) Beyond The Greenbelt: Extended Urbanization On The Shores Of Lake Simcoe
  • Rajendran K (2021) Conservation and Protection of Peri-Urban Rural Landscapes from the Impacts of Urbanization: Case Study of Manimangalam, Mahanyam and Malaipattu Villages in Manimangalam Watershed. In: Thirumaran K, Balaji G, Prasad ND (eds) Sustainable Urban Architecture. Springer, Singapore, pp 17–34 cited
  • Englezos N., Kartala X., Koundouri P., Tsionas M. & Alamanos, A. (2021). A novel hydro-economic – econometric approach for integrated transboundary water management under uncertainty. DEOS Working Papers. http://wpa.deos.aueb.gr/docs/INTEGRATED.TRANSBOUNDARY.WATER.MANAGEMENT.pdf
  • Wang T, Wang L, Ning Z-Z (2020) Spatial pattern of tourist attractions and its influencing factors in China. Journal of Spatial Science 65:327–344. https://doi.org/10.1080/14498596.2018.1494058 -

Cited

 Added: There is also rapid increase on built-up areas and the decline on the coverage of vegetation around lakes (Fitawok et al. 2020) which resulted in significant decrease on the size of wetland and water body (Assefa et al. 2021). For instance, studies show that over the last 35 years, the water bodies and wetlands around Bahir Dar city of Ethiopia decreased by 75.71% (Assefa et al. 2021) and same effect observed in other cities of the country.

  1. The reference style does not seem the usual one of mdpi with numbering. Corrected

Reviewer 3 Report

The manuscript is structured in a comprehensible way, the state of the literature is presented, the research question is derived and justified in a comprehensible way. The argumentation is stringent, the language clear and concise. The method is introduced concisely but comprehensibly and the results are discussed appropriately.
After reading the abstract, I had the expectation that the unintended side effects of urbanization along water bodies would also be addressed more in terms of content (such as ecological). Here it is recommended to snchronize the abstract and the content of the text more.
The legend of the map in Figure 1 is not to be read. The colors are indistinguishable for persons with color vision impairment. In addition, this is an island map. Here, it is recommended to create a separate map that also takes into account the spatial context and uses an accessible color scale.
The cartograms in the mislabeled Figure 2 (line 329) are distorted.
The manuscript has numerous formatting errors:
- It is not cited according to MDPI standards (and there are also errors in the citation style chosen: In line 87 is found the abbreviated first name in the citation).
- The text should be checked for multiple consecutive spaces (for example, lines 84, 94, 96).
- In line 108, the font size is partially inconsistent. Line 177ff is set in bold italics.
- In Figure 2 (which is erroneously labeled Figure 1), naming the numbers in the figure would be reader-friendly; also, an "s" is missing from "Small town...".
- In line 345, Table 2 is introduced without there having been a Table 1.

Author Response

After reading the abstract, I had the expectation that the unintended side effects of urbanization along water bodies would also be addressed more in terms of content (such as ecological). Here it is recommended to snchronize the abstract and the content of the text more.

The legend of the map in Figure 1 is not to be read. The colors are indistinguishable for persons with color vision impairment. In addition, this is an island map. Here, it is recommended to create a separate map that also takes into account the spatial context and uses an accessible color scale.

Figure 1 modified with visible lagend

The cartograms in the mislabeled Figure 2 (line 329) are distorted.

Corrected
The manuscript has numerous formatting errors:
- It is not cited according to MDPI standards (and there are also errors in the citation style chosen: In line 87 is found the abbreviated first name in the citation).

Corrected as Krugman (1994) and the entire citation changed into the numbered format as per the guideline of the MDPI
- The text should be checked for multiple consecutive spaces (for example, lines 84, 94, 96).

Addressed: line 84 deleted.
- In line 108, the font size is partially inconsistent. Line 177ff is set in bold italics.

The font size of line 108 and the bold italic on paragraph 177…corrected to be uniform with the reset of the texts.

- In Figure 2 (which is erroneously labeled Figure 1), naming the numbers in the figure would be reader-friendly; also, an "s" is missing from "Small town...".

Number of the figure corrected as ‘2’ and the missing ‘s’ added

- In line 345, Table 2 is introduced without there having been a Table 1.

Numbers of the tables changed from 2,3,4,5,6 to 1,2,3,4,5

Round 2

Reviewer 3 Report

Many thanks for the extensive revision. From my point of view, the article is now much more rounded and should be published.